# Osteoporosis: Molecular Pathology, Diagnostics, and Therapeutics

**DOI:** 10.3390/ijms241914583

**Published:** 2023-09-26

**Authors:** Babapelumi Adejuyigbe, Julie Kallini, Daniel Chiou, Jennifer R. Kallini

**Affiliations:** 1David Geffen School of Medicine, The University of California, Los Angeles (UCLA), Los Angeles, CA 90095, USA; badejuyigbe@mednet.ucla.edu; 2Department of Computer Science, Stanford University, Stanford, CA 94305, USA; kallini@stanford.edu; 3Department of Orthopedic Surgery, The University of California, Los Angeles (UCLA), Los Angeles, CA 90095, USA; danielchiou@mednet.ucla.edu

**Keywords:** osteoporosis, pathophysiology, models, hormones, cytokines, hormones, bone homeostasis, cellular mechanisms

## Abstract

Osteoporosis is a major public health concern affecting millions of people worldwide and resulting in significant economic costs. The condition is characterized by changes in bone homeostasis, which lead to reduced bone mass, impaired bone quality, and an increased risk of fractures. The pathophysiology of osteoporosis is complex and multifactorial, involving imbalances in hormones, cytokines, and growth factors. Understanding the cellular and molecular mechanisms underlying osteoporosis is essential for appropriate diagnosis and management of the condition. This paper provides a comprehensive review of the normal cellular and molecular mechanisms of bone homeostasis, followed by an in-depth discussion of the proposed pathophysiology of osteoporosis through the osteoimmunological, gut microbiome, and cellular senescence models. Furthermore, the diagnostic tools used to assess osteoporosis, including bone mineral density measurements, biochemical markers of bone turnover, and diagnostic imaging modalities, are also discussed. Finally, both the current pharmacological and non-pharmacological treatment algorithms and management options for osteoporosis, including an exploration of the management of osteoporotic fragility fractures, are highlighted. This review reveals the need for further research to fully elucidate the molecular mechanisms underlying the condition and to develop more effective therapeutic strategies.

## 1. Introduction

Osteoporosis is a very common condition affecting over 14 million people in the United States (US) and over 200 million people globally [1,2]. It is estimated that one in three women and one in five men aged 50 or over will suffer osteoporosis-related fragility fractures [2]. This heavy disease burden translates to staggering economic costs. The current annual economic burden due to osteoporosis is USD 6.5 trillion between the US, Canada, and Europe alone—and this figure is rapidly growing [3]. The annual US economic burden is projected to climb to USD 25.3 billion by 2025 [3].

Osteoporosis is a multifactorial condition characterized by changes in bone homeostasis, which result in reduced bone mass, impaired bone quality, and an increased propensity for fractures [1,4]. Hormones, cytokines, and growth factors regulate bone homeostasis both directly and indirectly. Peak bone mass is said to be achieved when all these factors are working effectively in conjunction with one another. Thus, it is thought that imbalances in these molecular and cellular processes alter bone homeostasis, driving the pathophysiology of osteoporosis [5,6]. Other factors such as race, gender, behavior, and diet can also have influences on bone mass and the tendency to develop osteoporosis. As medical advancements continue to lengthen life expectancy, osteoporosis has emerged as a major public health concern. Thus, understanding its cellular and molecular pathophysiology is crucial for appropriate diagnosis and management. In this paper, the (1) normal cellular and molecular mechanisms of bone homeostasis are discussed, followed by a discussion of the disease state that is osteoporosis, outlining the (2) proposed pathophysiologic mechanisms, (3) diagnostic tools, and (4) current treatment algorithms of this prevalent condition. The goal is to provide a structured up-to-date review on the current understanding of osteoporosis.

## 2. Cellular and Molecular Mechanisms

### 2.1. Structural and Cellular Components of Bone

Bone is a dynamic, mineralized, multifunctional connective tissue with organic and inorganic components. The organic component of bone is commonly referred to as the “osteoid” and is composed of both collagenous (mainly collagen type I) and non-collagenous (glycosaminoglycans and glycoproteins) proteins [7,8]. Each of these components—together with hormones, cytokines, and the cellular components of bone—regulate bone metabolism, deposition, mineralization, and turnover. The inorganic component of bone consists mainly of calcium and phosphorus hydroxyapatite crystals, which provide chemical rigidity and structure to the bone and account for 50–70% of bone mass [7,8,9].

The remaining bone volume can be attributed to its cellular components, chiefly composed of osteocytes, osteoblasts, and osteoclasts. Each of these cell types plays a dynamic role in the creation and maintenance of bone integrity. Osteocytes are found in the lacunae of the matrix and have a mechano-sensory function, maintaining homeostasis through the transmission of mechanical forces into chemical signaling pathways using various signaling molecules and proteins [5]. Osteoblasts are derived from undifferentiated mesenchymal cells and primarily function in bone formation, growth, and maintenance [5]. These multinucleated giant cells are synchronized by the chemical signaling pathways regulated by osteocytes [5]. Finally, osteoclasts are multinucleated giant cells primarily responsible for bone resorption. They are produced by the fusion of hematopoietic stem cells derived from monocytic precursors. They function primarily to resorb bone, preparing the osteoid matrix for bone formation [5,6].

The cellular, organic, and inorganic components of bone are arranged in specific microstructural units, termed osteons, which are composed of a harversian canal, lamellae, lacunae, and canaliculi, all arranged in a concentric pattern [10]. Macroscopically, bone is further organized into distinct structures, giving rise to two major types of bone within the adult skeleton: cortical and trabecular bone. Cortical bone comprises approximately 80% of the adult bone mass, whereas trabecular bone makes up the remaining 20%. Cortical bone is dense and has a relatively low turnover rate of 3%. It functions mainly to maintain the multiaxial strength and integrity of the bone. In contrast, trabecular bone is highly porous, with a relatively high turnover rate of 26%. It is more metabolically active than cortical bone.

### 2.2. Bone Homeostasis

The human skeletal system is a specialized, dynamic organ that requires continuous remodeling to maintain its structural and mechanical integrity. Bone remodeling begins in early fetal life and constantly functions thereon to strengthen and replenish the skeletal system as it sustains physical loads. Remodeling is a complex yet coordinated process that involves the aforementioned cell types, and the organic and inorganic components of bone [10,11,12]. Additionally, there are various proteins and signaling molecules that are also involved and act to further regulate bone homeostasis [4]. Impairment in this process may lead to mechanical and structural bony pathologies, including osteoporosis [11,12]. Bone remodeling can be separated into five phases: (1) activation; (2) resorption; (3) reversal; (4) formation; and (5) termination. A summary of the most important aspects of each of these phases is detailed in Table 1 below.

In the (1) activation phase, bone remodeling is initiated by local mechanical or systemic hormonal signals. During this phase, local (TGF-β, macrophage colony-stimulating factor (M-CSF), and receptor activator of NF-κB ligand (RANKL)) and systemic (vitamin D, calcium, parathyroid hormones (PTHs), estrogen, androgen, and glucocorticoids) regulators and transcription factors promote resorptive osteoclastogenesis. RANKL interacts with the RANK receptor (forming the RANKL-RANK complex) on osteoclast precursors, potently inducing differentiation into multinucleated osteoclasts. The osteoblast expression of M-CSF also promotes osteoclast survival and maturation. Additionally, during this stage, osteoblasts release chemokines to recruit osteoclast precursors and matrix metalloproteinases (MMPs) to further prepare the bone surface for remodeling [4,6,7].

During the (2) resorption phase, mature osteoclasts secrete MMPs to digest both mineral and organic bone matrices. This process involves the creation of Howship’s resorption lacunae, which are small spaces or pits in the bone. These lacunae are covered by canopy cells—flattened cells covering the surface of the bone. The size and shape of the lacunae are indicative of the activity of osteoclasts and the degree of bone resorption. Osteoprotegerin (OPG) can block RANK-RANKL complex formation and reduce resorption by inhibiting osteoclast differentiation and promoting apoptosis [4,6,7].

The (3) reversal phase is responsible for the crucial coupling of osteoclastic and osteoblastic activity at the site of remodeling. It begins with the apoptosis of mature osteoclasts. Osteoblasts are then directed to the resorption site in preparation for bone formation. Local molecules such as TGF-β play a pivotal role in attracting and preparing osteoblasts to initiate bone formation [4,6,7].

In the (4) formation phase, local and systemic regulators, such as Wnt, sclerostin, and PTH, induce osteoblastogenesis in bone. During this phase, osteoblasts deposit unmineralized osteoid until the area of previously resorbed bone is replaced. Bone formation is then completed as osteoid is gradually mineralized through the incorporation of hydroxyapatite. The balance between sclerostin, Wnt, and PTH is essential in bone formation. At rest, osteocytes express sclerostin, which prevents Wnt signaling (an inducer of bone formation) in osteoblasts. However, during bone formation, sclerostin expression is inhibited by PTH or mechanical stress, which allows for Wnt-induced bone formation to progress [4,6,7].

In the (5) termination phase, the rate of bone formation and bone resorption equivocates, and the remodeling cycle is terminated. The process of termination is completed through a series of yet undetermined termination signals. Bone mineralization also continues during this phase [4,6,9].

### 2.3. Molecular and Local Regulation

Bone remodeling is governed by both hormonal/chemical and mechanical signals. Systemic regulators of bone include estrogen, growth hormone, thyroid hormones, glucocorticoids, and androgens.

Thyroid hormones are essential for normal musculoskeletal development, maturation, metabolism, structure, and strength, as they promote bone turnover by influencing osteoblast and osteoclast activities. Glucocorticoids prolong osteoclast survival and reduce bone formation by increasing osteoblast apoptosis. At high doses, PTH increases bone resorption indirectly by promoting RANKL/M-CSF expression and inhibiting OPG expression [7]. At lower doses, PTH induces bone formation by promoting an increased survival, proliferation, and differentiation of osteoblasts. Other systemic regulators include vitamin D3, calcitonin, insulin-like growth factor (IGF), prostaglandins, and bone morphogenetic proteins [7].

Local regulators of bone remodeling include cytokines, growth factors, sirtuins, protein kinases such as the mechanistic target of rapamycin (mTOR), forkhead proteins, M-CSF, Wnt, sclerostin, and the RANK/RANKL/OPG system. Each of these signaling molecules plays a different role in the phases of bone remodeling [7]. Sirtuins inhibit sclerostin activity to promote Wnt signaling and bone formation. The increased activity of mTOR translates to increased osteoclastic activity and the release of cathepsin K. The microenvironment within bone is such that all these systemic and local regulators are delicately balanced and tightly regulated. Altered intracellular signaling milieus lead to pathological outcomes [7].

## 3. Osteoporosis Pathophysiology

Osteoporosis is a disorder characterized by decreased bone mass, density, quality, and strength, as shown in Figure 1. It is caused by imbalances in the process of bone remodeling to favor MSC senescence—and a shift in differentiation potential to favor adipogenesis over osteogenesis. In this pathological state, bone loses its structural integrity and becomes more susceptible to fractures [13]. This imbalance is primarily linked to variations in the activity levels of osteoclasts and osteoblasts.

Osteoporosis can be classified into two major groups: primary and secondary osteoporosis. Primary osteoporosis includes conditions for which there is no underlying medical etiology. These include idiopathic and involutional osteoporosis. Idiopathic osteoporosis occurs mostly in children and young adults and continues to have no known etiopathogenesis [6,14]. Involutional osteoporosis affects both men and women and is known to be closely related to aging and hormonal imbalances. Involutional osteoporosis can be further classified into Type I and Type II. Type I involutional osteoporosis mostly affects postmenopausal women and is often referred to as “postmenopausal osteoporosis”. This condition affects women between 51 and 71 years of age and is characterized by rapid bone loss [6,14]. Type II involutional osteoporosis—often referred to as “senile osteoporosis”—mostly affects those above 75 years of age. This condition is characterized by a primarily trabecular and cortical pattern of bone loss [6,14]. Secondary osteoporosis occurs due to an underlying disease or medication use, and it accounts for less than 5% of all cases of osteoporosis [6,14].

Traditional pathophysiological models of osteoporosis have emphasized the endocrine etiology of the condition. Estrogen deficiencies and the resultant secondary hyperparathyroidism, as described by models, coupled with an inadequate vitamin D and calcium intake, have been touted as the key determinants in the development of osteoporosis [15]. The postmenopausal cessation of ovarian function and subsequent decreases in estrogen levels have been known for decades to be key events in the acceleration of bone loss. The effects of estrogen loss are mediated by the direct modulation of osteogenic cellular lineages via the estrogen receptors on these cells. Specifically, decreased estrogen leads to simultaneous increases and decreases in osteoclast and osteoblast activities, respectively, leading to metabolic imbalances favoring bone resorption. Similarly, it is known that nutritional imbalances, specifically in vitamin D and calcium, can also promote bone resorption [15,16].

However, emerging research on bone homeostasis suggests that the pathophysiological mechanisms of osteoporosis extend beyond this unilateral endocrine model [15]. Rather, more dynamic models are being explored as the pathophysiological drivers behind the disease.

An important discussion point is the use of animal models for osteoporosis. Because of the similarity in pathophysiologic responses between the human and rat skeleton, the rat is a valuable model for osteoporosis. Rats are safe to handle, accessible to experimental centers, and have low costs of acquisition and maintenance [17]. Through hormonal interventions, such as ovariectomy, orchidectomy, hypophysectomy, and parathyroidectomy), and immobilization and dietary manipulations, the laboratory rat has provided an aid to the development and understanding of the pathophysiology of osteoporosis [17].

### 3.1. Osteoimmunological Model

The osteoimmunological model is a relatively novel one that capitalizes on the interplay between the immune system and the skeletal system [15]. It has become increasingly clear that the immune and skeletal systems share multiple overlapping transcription factors, signaling factors, cytokines, and chemokines [15].

Osteoclasts were the first cells in the skeletal system discovered to serve immune functions [15]. Some of the first insights into osteoimmunological crosstalk were gained by Horton et al. (1972), who explored the interactions between immune cells and osteoclasts leading to musculoskeletal inflammatory diseases [18]. The authors found that, in the pathophysiology of rheumatoid arthritis, the stimulation of bone resorption by osteoclasts is exclusively mediated by Th17 cells, which produce IL-17 to stimulate RANKL expression [15,18,19].

The osteoimmunological pathophysiological framework for osteoporosis is further strengthened by studies conducted by Zhao et al. (2016), who showed that osteoporotic postmenopausal women express increased levels of proinflammatory cytokines (TNF, IL-1, IL-6, or IL-17) when compared to their non-osteoporotic counterparts [20]. Cline-Smith et al. (2020) further strengthened this model by demonstrating a relationship between the loss of estrogen and the promotion of low-grade T-cell-regulated inflammation [21]. Regulatory T cells (T_reg_) have also been found to have anti-osteoclastogenic effects within bone biology through the expression of the transcription factor FOXP3 [22]. Accordingly, Zaiss et al. (2010) found that the transfer of T_reg_ cells into T-cell-deficient mice was associated with increased bone mass and decreased osteoclast expression [15,22].

B cells also play a role in the pathophysiology of osteoporosis. Panach et al. (2017) showed that B cells produce small amounts of both RANKL and OPG and modulate the RANK/RANKL/OPG axis [15,23,24].

### 3.2. Gut Microbiome Model

Another rapidly expanding model for the pathophysiology of osteoporosis explores the influence of the gut microbiome (GM) on bone health. It is now widely accepted that the GM influences the development and homeostasis of both the gastrointestinal (GI) tract and extra-GI tissues. GM health also affects nutrient production, host growth, and immune homeostasis [25,26,27]. Moreover, complex diseases, such as diabetes mellitus (DM), transient ischemic attacks, and rheumatoid arthritis, have all been linked to changes in the GM [25,26,27].

Ding et al. (2019) showed that germ-free mice exhibit increased bone mass, suggesting that a correlation exists between bone homeostasis and the GM [26]. This correlation was also redemonstrated by Behera et al. (2020)’s findings that the modulation of the GM through probiotics and antibiotics affects bone health [25,26]. Though the relationship between the GM and bone health is still being explored, various mechanisms have been proposed to explain this close “microbiota–skeletal” axis [15].

One such mechanism stems from the relationship between the GM and metabolism. The GM has been shown to influence the absorption of nutrients required for skeletal development (i.e., calcium), thereby affecting bone mineral density [28]. Additionally, nutrient absorption is thought to be influenced by GI acidity, which is directly regulated by the GM [15]. Moreover, the microbial fermentation of dietary fiber to short-chain fatty acids (SCFAs) also plays an important role in the regulation of nutrient absorption in the GI tract. Whisner et al. (2016) and Zaiss et al. (2019) recently reported that the consumption of different prebiotic diets (that can be fermented to SCFAs) was associated with an increased GI absorption of dietary calcium [29,30]. Beyond their influence on the GI tract, SCFAs have emerged as potent regulators of osteoclast activity and bone metabolism [31]. SCFAs have protective effects against the loss of bone mass by inhibiting osteoclast differentiation and bone resorption [15,32]. SCFAs are amongst the first examples of gut-derived microbial metabolites that diffuse into systemic circulation to affect bone homeostasis [15].

The GM also modulates immune functions. It is believed that the GM’s effect on intestinal and systemic immune responses, which, in turn, modulate bone homeostasis as described above, is yet another link between the GM and the skeletal system. Bone-active cytokines are released directly by immune cells in the gut, absorbed, and then circulate to the bone; these cytokines play a pivotal role in the GM–immune–bone axis [15,27].

Finally, it is understood that the bone-forming effect of intermittent PTH signaling closely depends on SCFAs—specifically, butyrate, a product of the GM. Li et al. (2020) provided evidence for butyrate acting in concert with PTH to induce CD4+ T cells to differentiate into T_reg_ cells. Differentiated T_reg_ cells then stimulate the Wnt pathway, which is pivotal for bone formation and osteoblast differentiation, as discussed above [15,33]. Interventions that focus on probiotics and targeting the GM and its metabolic byproducts may be a potential future avenue for preventing and treating osteoporosis.

### 3.3. Cellular Senescence Model

Cellular senescence describes a cellular state induced by various stressors, characterized by irreversible cell cycle arrest and resistance to apoptosis [34]. Senescent cells produce excessive proinflammatory cytokines, chemokines, and extracellular matrix-degrading proteins, known as the senescence-associated secretory phenotype (SASP) proteins [35]. The number of senescent cells increases with aging and has been linked to the development of age-related diseases, such as DM, hypertension, atherosclerosis, and osteoporosis [36].

Farr et al. (2016) explains the role of cellular senescence in the development of osteoporosis [37]. These authors found that there is an accumulation of senescent B cells, T cells, myeloid cells, osteoprogenitors, osteoblasts, and osteocytes in bone biopsy samples from older, postmenopausal women compared to their younger, premenopausal counterparts, suggesting that these cells become senescent with age [37]. Further studies conducted by Farr et al. (2017) suggest a causal link between cellular senescence and age-related bone loss by showing that the elimination of senescent cells or the inhibition of their produced SASPs had a protective and preventative effect on age-related bone loss [38]. These findings suggest that targeting cellular senescence through “senolytic” and “senostatic” interventions may have good results.

### 3.4. Genetic Component of Osteoporosis

Bone mineral density has up to 80% of variance in twin studies and is a heritable trait. Single-nucleotide polymorphisms in specific genes, in addition to polygenic and multiple gene variants, have been identified [39]. Makitie et al. summarized up to 144 different genes that have been reported to be linked to variances in bone mineral density [39]. As an example, Zheng et al. showed that rs11692564 had an effect of +0.20 SD for lumbar spine BMD [40]. It is well known that the WNT pathway plays a role in bone homeostasis, as it promotes bone cell development, differentiation, and proliferation [41]. Dysregulation in its signaling pathway leads to changes in bone mass, such as osteoporosis pseudoglioma syndrome, Pyle’s disease, and van Buchem disease [39]. PLS3 is another recently identified gene that is linked to early-onset osteoporosis. PLS3 functions by altering osteocyte function through an abnormal cytoskeletal microarchitecture and bone mineralization [42,43]. Finally, there are several genes that have a known effect on changes in the bone extracellular matrix. COL1A1 and COL1A2 mutations are associated with osteogenesis imperfecta; XYLT2 leads to spondyloocular syndrome; and FKBP10 and PLOD2 mutations lead to Bruck syndrome 1 and 2, respectively [44,45].

## 4. Diagnosing Osteoporosis

Currently, the diagnosis of osteoporosis primarily relies on the assessment of bone mass through bone densitometry, also known as dual-energy X-ray absorptiometry (DEXA) [46]. The test uses low-dose X-rays to measure the density of bones, typically in the spine, hip, and wrist. An individual’s DEXA score is calculated based on the measured bone density values, and the probability of future fracture risk is thereupon determined [6,7].

Bone strength can be quantified using bone mineral density (BMD) and/or bone quality. While tools exist to accurately quantify BMD, the accurate measurement of bone quality within the clinical setting remains elusive. Thus, measurement of the BMD is the most effective method for determining the rate of bone loss and monitoring disease progression [47]. Bone mineral content (BMC), however, is the bone mineral density summed over a projected area [48]. Peak bone mass is the amount of bony tissue present at the end of skeletal maturation [49]. Men tend to have higher peak bone mass than women, and African-American males and females have a higher peak bone mass than their Caucasian counterparts [50].

The World Health Organization (WHO) Expert Committee classification of BMD values is as follows: (i) normal: BMD > −1 SD t-score; (ii) osteopenia: BMD between −1 SD and −2.5 SD t-score; (iii) osteoporosis: BMD < −2.5 SD t-score; and (iv) established osteoporosis: BMD < −2.5 SD t-score + fragility fracture [51,52]. For premenopausal women, men under 50 years of age, and children, the Z-score (in relation to normal subjects of the same age and sex) is considered, with “normal” being considered up to −2.0 [53]. This classification is widely accepted as a diagnostic criterion, with sensitivity and specificity close to 90% [6].

In addition to bone densitometry, general blood and urine tests can provide important information about an individual’s overall health and any underlying conditions that may be contributing to osteoporosis. These markers are particularly useful for identifying metabolic bone diseases, as they can provide information not directly obtained through bone density measurements [6]. If ancillary testing is indicated, then an array of bone turnover markers (BTMs) can be measured. BTM testing detects peptides produced during bone matrix formation and degradation. Examples of bone formation markers include alkaline phosphatase (ALP) and osteocalcin (OC), which quantify osteoblastic activity. Of note, ALP has low sensitivity and specificity in metabolic bone disorders since it is secreted by various tissues, including the liver, bone, and placenta [54,55]. In contrast, there are BTMs specific to bone resorption. Degradation markers, such as pyridinoline (Pir) and deoxypyridinoline (Dpir), are proxy measurements for osteoclast activity. The most often measured resorption markers in clinical practice for the diagnosis of osteoporosis are the C-terminal telopeptide of type I collagen (ICTP), β-CrossLaps (β-CTX), and the N-terminal telopeptide of type I collagen (NTX) [56].

Several algorithms have been developed to estimate a patient’s future fracture risk [14]. The most commonly used algorithm in the US is the Fracture Risk Assessment Tool (FRAX). The FRAX and other such algorithms allow for the estimation of the 10-year risk of major osteoporotic fragility fractures (vertebral, hip, distal radius, or proximal humerus) [14]. The clinical risk factors that are utilized in these predictive algorithms include age, sex, prior history of osteoporotic fracture, femoral neck BMD, body mass index (BMI), glucocorticoid use, parental history of hip fracture, secondary causes of osteoporosis, smoking history, and alcohol consumption [14].

### Novel Diagnostic Approaches

In a recent study, the assessment of BMD using Hounsfield unit (HU) measurements from computed tomography (CT) scans was correlated with DEXA scan results [57]. This study established that glenoid and proximal humerus HU can reliably be measured and correlated with patients’ DEXA [57].

Earp et al. (2021) further concluded that the utilization of opportunistic HU values obtained from shoulder CT scans obtained for other purposes could assist in the earlier detection of abnormal bone density, offering an additional way to identify patients who may benefit from further diagnostic testing and potential treatment [57]. This study was the first of its kind and shows incredible promise for the novel diagnostic approach to osteoporosis using CT [57].

## 5. Treatment Options

Osteoporosis has been dubbed “the silent killer of the 21st century”, as there are minimal clinical signs of the condition prior to patients suffering fracture [14]. The rapidly evolving understanding of the pathophysiology of osteoporosis has led to diagnostic and therapeutic advances. The primary goal of most treatment options for osteoporosis is to reduce the risk of fractures and the subsequent associated morbidity and mortality [6]. The management of osteoporosis includes both non-pharmacological and pharmacological approaches [15].

### 5.1. Non-Pharmacological Treatment Options

The non-pharmacological management of osteoporosis includes various lifestyle and dietary interventions, which aim to upregulate bone production and inhibit bone resorption [58,59,60]. It has long been established that regular weight-bearing exercise stimulates bone production, increases bone strength, and is protective against fractures [58,59,60]. Children and young adults who are consistently active reach higher peak bone masses than those who are not [61,62]. In their systematic review, Howe et al. (2011) found that the most effective type of exercise for increasing femoral neck BMD was “high-force” exercise such as progressive resistance training [63]. LeBoff et al. (2022) highlighted the importance of weight-bearing exercises (in which bones and muscle work against gravity with feet and legs bearing body weight) and detailed the importance of a “multicomponent program” to adequately strengthen bone in patients with osteoporosis [49,50,51,52,53,54,55,56,57,58,59,60,61,62,63,64]. A multicomponent program should include progressive resistance training, balance training, back extensor strengthening, core stabilizers, cardiovascular conditioning, and impact or ground-reaction forces to stimulate bone [64].

Smoking has been shown to influence bone health indirectly and directly. Animal studies have shown that exposure to smoking can change the ratio of RANKL/OPG and lower levels of OPG, thus influencing osteoclast function [65,66]. Cheraghi et. al’s meta-analysis demonstrated that persons consuming 1–2 drinks daily had a 1.34 times increased risk of developing osteoporosis, and those who drank more than two drinks daily had a 1.63 times increased risk of developing osteoporosis [67]. This is hypothesized to be secondary to decreased bone remodeling due to lower levels of osteocalcin and C-telopeptide of type 1 bone collagen [67]. Therefore, smoking cessation should be considered as a non-pharmacological intervention to address osteoporosis.

The Canadian Multicenter Osteoporosis Study showed that an increased intake of protein and nutrient-dense foods, such as fruits, vegetables, and whole grains, was associated with a lower fracture risk [68]. Similar trends were seen in different diets among different cultures. Asian diets, which are low in dairy products, have shown that an increased consumption of large quantities of dark green vegetables may provide an adequate daily calcium dose [69]. When compared to an omnivore diet, a vegan diet had a higher prevalence of vitamin D insufficiency, although bone loss was comparable after 2 years between these two groups [70]. The Framingham heart study showed that, among men, a diet high in fruit, vegetables, and cereal was associated with a higher bone density [71].

### 5.2. Pharmacological Treatment Options

Before beginning pharmacological treatment for osteoporosis, individuals should be assessed to identify any secondary causes of the condition. If identified, these secondary causes should be addressed in concert with the osteoporosis. When beginning osteoporosis pharmacological therapy, it is also imperative to monitor BTMs to ensure the effectiveness of the treatment regimen [72].

There are several pharmacological options available for treating osteoporosis: (1) calcium and vitamin D supplementation; (2) antiresorptive agents (i.e., bisphosphonates and denosumab); (3) hormonal agents (i.e., estrogen, testosterone, and PTH analogues); and (4) novel therapies (romosozumab and Dickkopf-1 (Dkk1) inhibitors) [7].

#### 5.2.1. Calcium and Vitamin D Supplementation

In many cases of osteoporosis, dietary sources of calcium and vitamin D are inadequate. Additionally, the natural, physiological processes of aging systemically affect the body’s ability to naturally absorb calcium and vitamin D. Thus, it is recommended for those with, or at risk of developing, osteoporosis to supplement with additional vitamin D and calcium. Recent studies have found that calcium and vitamin D supplementation reduced the risk of hip fracture by 30% and the total fracture risk by 12–15% [7,73,74]. At least 700 International Units (IU) of vitamin D is needed for improving physical function and preventing falls and fractures. Supplementing calcium for a maximum total daily calcium intake of 1000 to 1200 mg has also been recommended. Even patients with osteoporosis on vitamin D supplementation should have regular lab work to ensure that 25-hydroxy vitamin D levels of more than 50 nmol/L are maintained.

#### 5.2.2. Antiresorptive Agents

##### Bisphosphonates

Bisphosphonates are a type of medication that strongly binds to hydroxyapatite, inhibiting osteoclast-mediated bone resorption and increasing BMD [7]. Multiple studies have well established that bisphosphonates reduce the risk of fractures in a wide range of patients, including those who are extremely frail [75,76].

In spite of its clinical benefits, bisphosphonate use has also been associated with a number of adverse effects, which include gastrointestinal symptoms, bone/joint pain, esophageal ulceration, and, rarely, osteonecrosis of the jaw (the highest risk of which is in patients with cancer) [7]. The prolonged use of bisphosphonates (5+ years) has also been associated with an increased risk of atypical femur fractures [7,77]. Given this risk, it is imperative to evaluate individuals on prolonged bisphosphonate treatment regimens on an individual basis, with drug holidays and alternative treatment options considered following use for 5+ years [77].

##### Denosumab

Denosumab is a humanized monoclonal antibody that decreases osteoclastic activity by inhibiting RANKL [7]. The 2011 international, randomized, placebo-controlled Fracture Reduction Evaluation of Denosumab (FREEDOM) study showed a reduction in fracture incidence of 68% for vertebral fractures, 40% for hip fractures, and 20% for non-vertebral fractures in the first three years in postmenopausal woman taking denosumab [78]. Denosumab is used as an alternative to bisphosphonates when they are not tolerated or are contraindicated.

Treatment with denosumab is usually 5–10 years in duration, after which the antiresorptive effects rapidly decrease. Consequently, atypical fracture risk increases in a manner similar to the prolonged bisphosphonate risk [7]. Other adverse effects include hypocalcemia, skin rash, an increased risk of bacterial infections, and osteonecrosis of the jaw.

#### 5.2.3. Hormonal Agents

##### Estrogen and Selective Estrogen Receptor Modulators (SERMs)

Estrogen regulates bone remodeling by blocking RANKL and by increasing OPG production by binding to the ERα receptor—a receptor mainly found in bones. In normal conditions, estrogen’s inhibitory effect on osteoclast activity helps maintain a balance in the bone remodeling process. However, prolonged estrogen treatment can cause serious side effects, such as breast cancer, deep vein thrombosis (DVT), and stroke [79].

To mitigate these issues, SERMs (such as raloxifene and lasoxifene) were developed to provide the benefits of estrogen while minimizing the associated adverse effects [7]. They are mainly used for treating and preventing osteoporosis in postmenopausal women after first-line options have been exhausted [4,7]. They have been shown to be particularly effective in decreasing vertebral fracture risk, though they do decrease the risk of all fragility fractures to some extent [7].

The adverse effects from the prolonged use of SERMs are similar to those associated with estrogen use—namely, an increased risk of breast cancer, DVT, and stroke—though they occur much more rarely than with estrogen [7]. Additionally, the sudden discontinuation of SERMs following prolonged use can result in rebound increases in bone remodeling, which can, in turn, lead to increased bone loss. Thus, when treatment is discontinued, patients should transition to another treatment agent immediately [7,80].

##### PTH Analogues

Just as low, pulsatile doses of PTH stimulate bone growth, so, too, does the timed administration of PTH analogues such as teriparatide. Teriparatide is a synthetic version of PTH, and it functions as an anabolic agent in bone (as opposed to antiresorptive agents, which have been previously discussed). Recent studies have shown that PTH analogues effectively increase BMD and lower the risk of vertebral fractures. These agents are used as another treatment option when first-line therapies fail [7,81].

It is contraindicated to use these medications in patients with Paget’s disease, skeletal muscle metastases, or previous bone radiation therapy. Additionally, the adverse effects from these therapies include nausea, myalgia, arthralgia, headache, and dizziness [7]. Prolonged, unregulated use is also associated with increased bone resorption. Thus, the use of PTH analogues should be restricted to a duration of two years [82].

#### 5.2.4. Novel Therapies

##### Romosozumab

Romosozumab is a newly approved monoclonal antibody targeting sclerostin. Approved by the Food and Drug Administration (FDA) in April 2019, it uniquely exhibits the ability to stimulate bone formation while simultaneously reducing bone resorption. It achieves this by upregulating the Wnt pathway while also acting as an enhancer of RANKL synthesis, thereby downregulating this latter pathway [83,84,85]. Romosozumab is a potent treatment option, both alone and in combination with other drugs [86]. Currently, it exists in an injectable form and is recommended for women without a high risk of cardiovascular disease.

However, its anabolic effect is temporary and tends to wear off [87]. As of 2021, it has reached Phase III trials, which have shown an increase in bone mineral density and a decrease in vertebral and hip fractures. These were based on two studies: the Fracture Study in Postmenopausal Women with Osteoporosis (FRAME trial) and the Active-Controlled Fracture Study in Postmenopausal Women with Osteoporosis at High Risk (ARCH study) [88,89]. In the FRAME study, romosozumab use showed a 73% reduction in new vertebral fractures compared with a placebo [88]. The ARCH study showed a 48% lower risk of new vertebral fractures in the group that received romosozumab and alendronate than in the alendronate-only group [89]. However, major cardiac events were observed in the ARCH study, while headaches, arthralgia, and injection site reactions were also observed [89,90]. Romosozumab has an estimated cost of USD 1825 a month, which is similar in cost to denosumab and conjugated drugs [90].

### 5.3. Orthopedic Management of Fragility Fractures

The therapeutic principles of osteoporotic fracture include fracture reduction, immobilization, physical therapy, and anti-osteoporosis treatment [91]. Management involves a combination of all four principles to facilitate the most optimal outcomes. Reduction procedures should be performed carefully to avoid further harm and to allow for early mobilization and rehabilitation once the fracture is stabilized. Anti-osteoporosis treatment is also important to prevent worsening of the underlying osteoporosis and fracture-related complications [91].

The treatment plan varies based on the patient’s specific fracture, degree of osteoporosis, and overall health. The focus should be on tissue repair and functional rehabilitation rather than anatomical fracture reduction [91].

For patients who require surgical intervention, orthopedic surgeons must keep in mind that fragility fractures tend to heal much more slowly than do traumatic fractures. To prevent complications, surgery should involve minimal trauma to the surrounding tissues and aim to best restore the articular surface if a fracture extends into the joint [91].

Addressing Common Fragility fractures:Vertebral Fractures:

The most common osteoporotic fractures occur within the vertebral column, with 85% of patients experiencing some level of pain and the remaining 15% being asymptomatic [92]. In cases of mild midline back or paraspinal pain, no neurological deficits, and minimal vertebral compression (less than one-third vertebral height loss), non-surgical treatment is recommended. Minimally invasive surgery is preferred for patients with neurological deficits, severe vertebral compression (more than one-third vertebral height loss), damage to the posterior vertebral wall, and significant pain that does not respond to conservative treatment [91].

Hip Fractures:

Hip osteoporotic fractures primarily occur in the femoral neck and intertrochanteric area and are marked by high rates of deformity, disability, delayed recovery, and elevated mortality. Regarding femoral neck fractures, treatment options may include non-surgical or surgical methods depending on the patient’s individual characteristics and goals of care. Most US orthopedic surgeons manage femur fractures operatively if the patient and family are amenable, though this is less so the case in Europe. For minimally displaced or impacted fractures in patients with extremely poor health, non-surgical treatments, such as bed rest with weighted traction, brace immobilization, and nutritional support, may be considered [91]. Surgical options for femoral neck fractures include external or internal fixation, hemiarthroplasty, total hip arthroplasty, or proximal femoral replacement. Prompt surgical treatment within 24–48 h of injury has demonstrably improved patient morbidity and mortality [93,94].

Proximal Humerus Fractures:

For nondisplaced proximal humerus fractures, non-surgical treatment is the preferred option. This can involve the use of a sling or shoulder immobilizer. In cases of displaced fractures in a highly functional patient, surgical management should be considered and can involve open reduction and internal fixation (ORIF) or prosthetic replacement [91].

Distal Radius Fractures:

Osteoporotic fractures of the distal radius are often comminuted and can involve the articular surface, leading to deformities and chronic pain. Initial treatment should be aimed at closed manual reduction and casting/splinting, ensuring proper restoration and alignment of the articular surface and normal positioning of the wrist. In cases of unstable fractures or an inadequate manual reduction, ORIF may be required to more precisely restore the articular surface [91].

Atypical Femur Fractures:

Atypical femoral shaft fractures occur from the prolonged use of resorptive agents, some of which have been mentioned above (bisphosphonates, denosumab, and some SERMs) [95]. The prolonged use of these substances alters the balance between bone resorption and bone formation, favoring bone formation at first but then shifting over time to favor bone resorption. Ideally, the medical treatment courses that include bisphosphonates should not exceed five years [15,95]. The management of an atypical femur fracture is similar to that of a hip fracture; surgery within 24–48 h is recommended in elderly patients [95].

However, controversy exists on how to manage the contralateral, nonfractured femur in a patient who has sustained an atypical femur fracture due to the prolonged use of antiresorptive agents. Imaging studies are recommended for users who present symptomatically with hip, thigh, or groin pain. Conventional radiography, CT, DEXA, and MRI are all modalities that should be explored in these scenarios [95].

In individuals with an atypical femur fracture who have been treated with bisphosphonates, immediate discontinuation of the medication is advised. Supplemental calcium and vitamin D should be provided as needed. For patients with incomplete fractures and persistent pain for three months despite medical management, prophylactic intramedullary surgical nail fixation is recommended to prevent complete fractures [95,96]. Pharmacologically, interventions to promote bone healing and formation should also be considered. Teriparatide has been shown to promote fracture healing, even in cases of nonunion [97]. In a retrospective case–control study, Miyakoshi et al. (2015) observed a reduction in healing time and an increased union rate with the use of teriparatide [98]. Surgical management through intramedullary nailing or plating is recommended for patients who sustain complete fractures [99].

However, controversy exists on how to manage the contralateral, nonfractured femur in a patient who has sustained an atypical femur fracture due to the prolonged use of antiresorptive agents. Atypical femur fractures affect the contralateral leg in 28% of cases, with time ranges between fractures ranging from one month to four years [99,100]. Thus, adequate study of the contralateral leg is mandatory, as recommended by the European Medicines Agency (EMA) and the FDA [101,102]. The assessment of the contralateral femur should be performed during the initial hospitalization, with the aim of promptly determining appropriate treatment or preventive measures for the contralateral fracture, as detailed in Table 2 below. An X-ray evaluation of the entire contralateral femur is recommended, even in the absence of prodromal pain [102]. CT, DEXA, and/or MRI may also be used if clinical suspicion is high and conventional radiographs are unrevealing [95].

## 6. Conclusions and Future Prospects

In summary, osteoporosis is a global health concern with significant associated morbidity, mortality, and economic costs. The cellular component of bone includes osteocytes, osteoblasts, and osteoclasts, each playing essential roles in bone integrity and remodeling. Through our understanding of the molecular and cellular components behind bone homeostasis, we have been able to better refine our diagnostic and therapeutic protocols surrounding osteoporosis.

Our understanding of the pathophysiology of the disease continues to evolve, involving an array of interconnected models. These models involve different pathophysiological thought processes, involving different components of the human body. The osteoimmunological, gut microbiome, and cellular senescence models provide valuable insights into the multifactorial nature of osteoporosis and highlight new possibilities for future therapeutic approaches. Targeting immune interactions, the gut microbiome, or cellular senescence could potentially lead to more effective treatments and preventive measures for osteoporosis, enhancing bone health and reducing fracture-susceptible populations.

The rudimentary diagnosis of osteoporosis primarily still relies on bone densitometry, specifically DEXA, to assess BMD and determine the probability of future fracture risk. While this method is currently the most effective method for monitoring disease progression, accurately measuring bone quality within the clinical setting remains challenging. Additionally, various blood and urine tests can provide valuable information about overall health and underlying conditions contributing to osteoporosis in conjunction with predictive algorithms that examine contributing environmental and behavioral factors (i.e., FRAX). Novel diagnostic approaches, such as using Hounsfield unit (HU) measurements from computed tomography (CT) scans, show promise in the early detection of abnormal bone density, offering an additional way to identify patients who may benefit from further diagnostic testing and treatment.

Treating osteoporosis largely requires a combinatory approach that involves both old and new ways to address the disease. Environmental and behavioral factors, like smoking, alcohol consumption, diet, and weight-bearing exercise, continue to form much of the non-pharmacological approach to the disease. Supplementation with calcium and vitamin D has continued to remain relevant pharmacologically. Anti-hormonal and antiresorptive pharmacological agents are more novel treatment options that have been developed; however, providers should be cautious in their prescription given the profile of some of their side effects. Novel therapies like romosozumab, a monoclonal antibody targeting sclerostin, show promise in stimulating bone formation while reducing bone resorption. This therapy capitalizes on the osteoimmunological and cellular senescent models for the pathophysiology of osteoporosis. Future pharmacological interventions should keep these models in mind when developing novel therapies.

Finally, the orthopedic management of fragility fractures involves a combination of fracture reduction, immobilization, physical therapy, and anti-osteoporosis treatment. Atypical femur fractures, associated with the prolonged use of certain medications, require careful management, and they may involve surgical intervention and discontinuation of medication. When fragility fractures are sustained, management should be tailored to the individual and type of fracture. Disease management should extend beyond the management of the affected limb/bone to include consideration of the contralateral limb/bone. Specifically, the potential for contralateral femoral fractures should be managed with great care. Though there is contention around the necessity of this evaluation, we hope that this paper sheds light on the importance of managing these fractures. Additionally, we are hopeful that the algorithm provided in Table 2 becomes a regular standard of care for physicians managing patients with osteoporosis. As we continue to learn more about this important disease, it is imperative that we constantly revise our diagnostic and management practices to optimize patient outcomes.

## Figures and Tables

**Figure 1 ijms-24-14583-f001:**
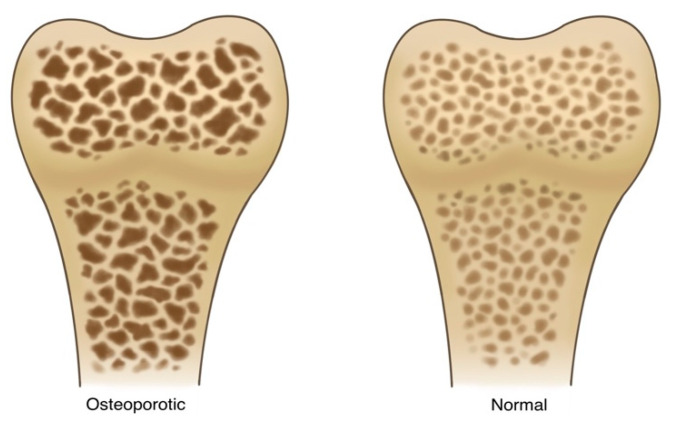
A side-by-side illustration depicting the stark contrast between normal and osteoporotic bone. The normal bone structure (**right**) is well defined, demonstrates a thick trabecular architecture within the bone matrix, demonstrates sufficient mineralization and calcium content, and, finally, shows minimal signs of fractures of degradation. The osteoporotic bone (**left**) shows signs of advanced bone loss and weakening. There is a dramatic reduction in trabecular density and thickness, resulting in a porous and fragile appearance. The pronounced gaps and voids within the bone matrix represent areas of compromised strength.

**Table 1 ijms-24-14583-t001:** Summary of phases of bone formation. MMPs = matrix metalloproteinases. OPG = osteoprotegerin. RANKL = receptor activator of nuclear factor kappa-Β ligand. TGF-β = transforming growth factor—beta.

**Activation Phase**	Local mechanical or systemic hormonal signals initiate bone remodeling and promote osteoclastogenesis. Osteoblasts release chemokines and MMPs to recruit osteoclast precursors and prepare bone surface for remodeling.
**Resorption Phase**	Mature osteoclasts continue to secrete MMPs to digest mineral and organic bone matrices. Howship’s resorption lucunae are created.
**Reversal Phase**	OPG can block RANK-RANKL complex formation and reduce resorption through apoptosis of mature osteoclasts. Osteoblasts directed to resorption site for bone formation by TGF-β
**Formation Phase**	Local and systemic regulators induce osteoblastogenesis. Osteoblasts then deposit unmineralized osteoid until the area of previously resorbed bone is replaced. Osteoid is gradually mineralized through incorporation of hydroxyapatite.
**Termination Phase**	Bone formation and resorption equilibrate, and the remodeling cycle is terminated. Bone mineralization continues.

**Table 2 ijms-24-14583-t002:** Summary algorithm for identification and management of atypical femur fractures in patients taking predisposing medications (i.e., bisphosphonates, denosumab, some selective estrogen receptor modulators). XR = X-ray. CT = computed tomography. DEXA = dual-energy X-ray absorptiometry. MRI = magnetic resonance imaging.

Management of Atypical Femur Fractures	
**Step 1:** **Identification**	Imaging recommended if hip, thigh, or groin pain Modalities: XR, CT, DEXA, MRI
**Step 2:** **Medical Management**	Immediate discontinuation of any offending medication Recommend Ca^2+^/Vit D supplementation Consider teriparatide supplementation
**Step 3:** **Surgical Management**	Strongly recommended for incomplete fractures and persistent pain for three months, unless contraindicated Strongly recommended for complete fractures, unless contraindicated
**Step 4:** **Management of Contralateral Femur**	Strongly recommend full contralateral femur imaging during initial hospitalization Modalities: XR, CT, DEXA, MRI

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
