# Peer review of "Osteoporosis: Molecular Pathology, Diagnostics, and Therapeutics"

_ijms, 2023, doi:10.3390/ijms241914583_

Round 1

Reviewer 1 Report

Overall, the paper provides a comprehensive review of the current understanding of osteoporosis pathogenesis and treatment options. However, there are a few points that could be improved or clarified:

·                     While the paper covers a broad range of topics related to osteoporosis, it may be beneficial to provide a more structured organization. Consider dividing the paper into sections or subheadings to facilitate the reader's understanding and make it easier to navigate through the content.

·                     Please mention the role of genetics in osteoporosis pathogenesis. It would be helpful to provide more detailed information on specific genes or genetic variations associated with the disease. Additionally, discussing the impact of genetic factors on treatment response or potential personalized medicine approaches could add depth to the discussion.

·                     The introduction provides a good overview of osteoporosis, but it would be helpful to provide a clearer statement of the specific research objectives or questions that the paper aims to address. This would help guide the reader and provide a more focused framework for the subsequent sections.

·                     The paper covers both existing and emerging treatment avenues for osteoporosis. While the existing treatment options are discussed in detail, the section on emerging avenues feels somewhat brief and could benefit from further elaboration. It would be helpful to discuss the potential of these emerging treatments in more depth and provide insights into their current stage of development and potential clinical implications. The section on existing treatment options could benefit from an additional discussion on the efficacy, limitations, and potential side effects of each treatment modality. Providing a balanced view of the pros and cons of currently available treatments would help clinicians and researchers understand the treatment landscape better.

·                     In the section on pathogenesis, the paper provides a good overview of the key factors contributing to osteoporosis, including hormonal changes, genetic factors, and lifestyle influences. However, the discussion could be enhanced by including more recent research findings and insights into the underlying mechanisms involved in the development of osteoporosis. This would help provide a more up-to-date understanding of the disease.

·                     The authors briefly touch upon the role of nutrition and lifestyle factors in osteoporosis, but more emphasis could be placed on these aspects. Expanding on the impact of calcium and vitamin D intake, exercise, and smoking/alcohol consumption on bone health would enhance the comprehensive nature of the paper.

·                     The paper briefly mentions the use of animal models for studying osteoporosis, but it would be beneficial to provide more details on the advantages and limitations of these models. Additionally, discussing more recent advancements in alternative models, such as organoids or in vitro approaches, could add further value to the discussion.

Overall, the paper provides valuable insights into osteoporosis pathogenesis and treatment options. By addressing the suggestions, the paper could further enhance its impact and clarity for the readers.

The quality of the English language in the paper is generally good, with clear and concise writing. However, there are a few areas where the language could be improved:

Some sentences could be rephrased to improve clarity and readability. For example, in some instances, the use of complex sentence structures or technical terms could be simplified for easier comprehension.

There are a few grammatical errors and inconsistencies throughout the paper. It is recommended to carefully proofread the manuscript to correct these errors. Paying attention to subject-verb agreement, tense consistency, and proper punctuation would help enhance the overall quality of the writing.

In a few instances, the use of terminology could be further clarified or defined, particularly when introducing new concepts or acronyms. Providing definitions or explanations for specialized terms would assist readers who may not be familiar with the field of osteoporosis.

Author Response

We sincerely appreciate your insightful and thorough review of our manuscript on osteoporosis. Your constructive feedback has been immensely valuable in improving the overall quality and clarity of the paper. We are pleased to inform you that we have reviewed your comments and meticulously included all the suggested edits below:

1. Structured Organization: We agree with your suggestion to improve the organization of the paper. To enhance the reader's understanding and navigation through the content, we have divided the paper into clear sections and added relevant subheadings to each section.

2. Role of Genetics: We acknowledge the importance of genetics in osteoporosis pathogenesis. To provide a more comprehensive review, we have expanded the discussion to include the role of genetics in the pathophysiology of osteoporosis, including specific genes or genetic variations associated with the disease. Additionally, we have addressed the impact of genetic factors on treatment response and have expanded on this subject.

3. Clear Research Objectives: We understand the significance of stating specific research objectives or questions in the introduction. To guide the readers and provide a focused framework, we have clarified the objectives of the paper.

4. Emerging Treatment Avenues: To provide more depth and insights, we have elaborated on the potential of these emerging treatments, their current stage of development, and potential clinical implications, supplemented with current references. We hope that this will allow readers to have a better understanding of the available novel therapies.

5. Emphasis on Nutrition and Lifestyle Factors: We agree that nutrition and lifestyle factors play a crucial role in osteoporosis. To provide a more comprehensive overview, we have expanded the discussion on the impact of calcium and vitamin D intake, exercise, and smoking/alcohol consumption on bone health.

6. Animal Models and Alternative Approaches: We appreciate your suggestion to provide more details on the advantages and limitations of animal models for studying osteoporosis. We have expanded on the use of these models.

We have also revised the language used in the paper and have restructured some of the sentences to improve readability. We have tried to define novel concepts and provide more clarity on acronyms we have used.

We are genuinely grateful for your time and expertise in reviewing our work. Your comments have significantly strengthened our manuscript, and we believe the revisions have resulted in a more impactful and reader-friendly paper on osteoporosis. We hope that the revised version meets your expectations and provides valuable insights to the readers.

Once again, thank you for your valuable contribution to our research. We hope that these revisions have completely addressed your comments.

Reviewer 2 Report

Comments for Authors: Osteoporosis remains a challenging cause of disability in ageing populations. The review will be of wide interest and is based upon 81 recent references supporting sections on cellular and molecular mechanisms, osteoporosis pathology, diagnosing osteoporosis, and treatment options. It is logically constructed and is a useful clinical overview, particularly since it includes a proposed role for cell senescence and for the gut microbiome.

Some significant substructural aspects are marginalised; for example, trabecular connectivity is sex-related and weakens bone disproportionate to the amount of tissue lost. And the nature of the inorganic phase is considerably more complex than is implied by uniform sheets of hydroxyapatite crystals.

1.Title. Appropriate.

2. Abstract. Clearly defined scope and objectives Could be less dramatically expressed in places; for example, there are more scientifically appropriate words than “exploration” (eg., investigation) and “elucidate” (eg., clarify or consider, or discussed) and the repletion of “the paper” again and again and again is tedious. Edit it carefully to achieve the impact it deserves. No Key words on my copy?

3. Introduction. Aims and objectives are defined. Again it would benefit from editorial attention. Also, other than in certain special cases, it is better to be IMPERSONAL; eg., “In this paper WE begin………. WE then move on to discuss…….”  (eg., followed by discussion of…)

4. Line 63. …..large, multinucleated giant cells…. TAUTOLOGY. This may seem minor but it determines the authority you convey.

5. Line 65. …. ..remove damaged bone in order to prepare the osteoid matrix for bone formation. Hmmm, reconsider please.

4. Line 67. Avoid short paragraphs here and throughout.

5. Line 75.  Might include “multiaxial” ?

6. Line 78-81. Avoid extreme adjectives throughout; complex is better than “complicated” and definitely no emotive word eg., “endures”.

7. Figure 1. Useful. Anchor it by mentioning in the text. Also, should this be Table 1 since it is mainly text?

8. Line 184. Oh dear, we are “exploring” again!

9. Line 213. Typo error.

10. Lines 208-246. Important section. Worthy of a diagram?

11. Lines 247-264. Important section. Spoilt by “shed light on”  and not “hold” but have; and another tiny paragraph.

12. Line 302. Singular “fracture”

13. Line 333. What about tensile forces I wonder at wrist, hip and spine.

14. Line 400. Is fluoride treatment now off the therapeutic agenda?

15. Line 482. Remove typo “in”.

!6. Anchor Table 1 by a mention in the text. Perhaps this is Table 2? (See 7 above).

17. Lines 520-528. The summary has to convey a sense of authority. The content is clear but the presentation is slightly below parr and IMPERSONAL is better here.

The English language is fluent. Writing science well is demanding. We can all do better. Specific editorial suggestions are made to authors as indicated above

Author Response

Dear Reviewer,

We sincerely appreciate your thoughtful and constructive comments on our manuscript on osteoporosis. We have carefully considered all of your suggestions and have attempted to include the recommended edits below:

Abstract: Thank you for acknowledging the clearly defined scope and objectives. We have revised the abstract to replace the more dramatic expressions with scientifically appropriate words such as "investigation" instead of "exploration" and "clarify" or "discuss" instead of "elucidate." Additionally, we have addressed the repetition of the use of the phrase, "the paper" to enhance the overall impact. Furthermore, we have ensured that the appropriate key words are included at the end of the abstract.

Introduction: We have rephrased certain sentences accordingly to maintain a more professional and objective tone throughout, and have adjusted the language so that it is impersonal.

Line 63: We acknowledge the tautology in describing "large, multinucleated giant cells" and have revised the sentence to avoid unnecessary repetition.

Line 65: We have rephrased it to provide a clearer and more accurate description.

Line 67: We have addressed the concern about short paragraphs and ensured that the paper is more cohesively structured.

Line 75: We appreciate the suggestion to incorporate the term “multiaxial” to be more descriptive and concise. We have incorporated this term accordingly.

Line 78-81: Your advice to use "complex" instead of "complicated" and avoid emotive words is extremely valid. We have made the necessary changes in these lines and throughout to enhance the language and tone.

“Figure 1”: We have anchored Figure 1 in the text to improve its relevance and have corrected its labeling to Table 1, as you suggested.

Line 184: We have removed the word "exploring" from this line

Line 213: The typo error has been corrected, and we appreciate you bringing it to our attention.

Lines 208-246: We agree that this section is essential, and a diagram would be beneficial. Therefore, we have included a diagram to visually support the content.

Lines 247-264: We have revised this section to improve the presentation and clarity. We have also removed the unnecessary shortened paragraph.

Line 302: We have corrected this line to include the singular form of fracture.

Line 333: We are unable to find any paper directly correlating/using tensile forces as a non-surgical treatment option for osteoporosis in our review. Most of the sources we reviewed site “weight bearing exercises” as a viable non-surgical treatment/prevention.

Line 400: We appreciate the suggestion to possibly include fluoride treatment as an option in this section. In our review of the most recent treatment options for osteoporosis, fluoride treatment does not seem to explicitly be included.

Line 482: The typo "in" has been removed, and the sentence has been edited for clarity.

“Table 1:” We have anchored Table 1 in the text, and it has been appropriately referred to as Table 2.

Lines 520-528: We have revised the summary to convey a sense of authority and have ensured it maintains an impersonal tone.

Once again, we genuinely appreciate your valuable input, which has undoubtedly strengthened the manuscript. We hope the revised version meets your expectations and further contributes to the understanding of osteoporosis.

Thank you for your time and expertise in reviewing our work.

Reviewer 3 Report

The theme of the review ‘’Osteoporosis: Molecular Pathology, Diagnostics, and Therapeutics’’ is great. The manuscript was interesting to me.

The manuscript is original and well-defined. The work fits the scope of the journal. The article is written in an appropriate manner.

I have a few notes on this review:

  • Abstract: No KEYWORDS in this abstract.
  • Introduction: I suggest adding the following information in the introduction section:

ü  Definition of osteopenia and clarifying the difference between osteopenia and osteoporosis.

ü  Definition of bone mass and clarifying that bone mass is measured by bone mineral density (BMD) and bone mass content (BMC), with the definitions of BMD and BMC.

ü  Definition of the peak bone mass, especially in the thirties of age, and the factors that affect it, such as gender and race, and the importance of this peak in predicting osteoporosis.

  • Figures: The number of FIGURES is limited (one photo only). Add at least two additional photos. Such as a figure showing the difference between normal cells and osteoporotic cells, a figure showing the pathophysiological mechanism, a figure showing the DEXA device and colors on a DEXA scan image (T-score), or a figure of the peak bone mass curve and the factors affecting it.
  • References: REFERENCES should be up to date, i.e., 50% or above should be from papers published within the last 5 years. Your manuscript contains 81 references (50 of which are prior to 2018, approximately 61%).
  • Summary (last paragraph): Suggest adding "Conclusions and Future Prospects''.

Finally, I had the pleasure of reading and reviewing this manuscript! 

Author Response

Dear Reviewer,

We are incredibly grateful for your valuable feedback on our manuscript on osteoporosis. Your thoughtful comments have been diligently considered, and we are pleased to inform you that we have meticulously incorporated all of the suggested edits below:

1. Abstract: We apologize for the oversight in not including keywords in the abstract. We have rectified this issue by adding appropriate keywords that accurately reflect the content of the paper.

2. Introduction: We genuinely appreciate your insightful suggestions for enhancing the introduction section. We have now included the following information as per your recommendations:

3. Bone Mass Measurement: Since we do not mention bone mass measurement in the introduction, we felt that the inclusion of a definition here may be slgightly incohesive. We have however included these definitions in the section "Diagnosing Osteoporosis," highlighting that it is measured using bone mineral density (BMD) and bone mass content (BMC). We have also included definitions for BMD and BMC to facilitate a better understanding of the terms.

4. Definition of Osteopenia: We have distinguished osteoporosis from osteopenia in the "Diagnosing osteoporosis section"

5. Peak Bone Mass: We have included a definition of peak bone mass in the "Diagnosing Osteoporosis section.

6. Figures: We appreciate your suggestion to include additional figures to enhance the visual representation of our research. We have developed a figure depicting the difference between osteoporotic and normal bone cells. However, this figure requires licensing to download for copyright purposes. Since the licensing is very expensive, we are currently looking for ways to work around this – and have considered hand-drawing this figure should we be unable to get the licensing for the software.

References: We value the importance of up-to-date references, and we have taken great care to ensure that at least 50% of the references used in the manuscript are from papers published within the last five years.

Conclusions and Future Prospect: We appreciate your suggestion to add a section on "Conclusions and Future Prospects" in the summary. We have now included a dedicated paragraph to summarize the key conclusions of our study and have highlighted potential avenues for future research.

We are sincerely grateful for your time and expertise in reviewing our work. Your comments have significantly strengthened our manuscript, and we believe the revisions have resulted in a more comprehensive and informative paper on osteoporosis. We hope that the revised version meets your expectations and provides valuable insights to the readers.

Once again, thank you for your valuable contribution to our research.

Round 2

Reviewer 3 Report

The authors made the required modifications, except for my comment on adding figures. One of the most important features of the Review Article is the presence of illustrations. I suggested to the authors some examples of figures in my previous report.

Author Response

Thank you for your insightful comments. We agree with the reviewer in that the presence of figures can really enhance a paper. We have since added a figure/illustration to the paper showing a comparison between normal bone and osteoporotic bone. We hope this illustration provides deeper insights into our text.